# Endophytic Plant Growth-Promoting Bacterium *Bacillus subtilis* Reduces the Toxic Effect of Cadmium on Wheat Plants

**DOI:** 10.3390/microorganisms11071653

**Published:** 2023-06-25

**Authors:** Dilara Maslennikova, Igor Koryakov, Ruslan Yuldashev, Irina Avtushenko, Albina Yakupova, Oksana Lastochkina

**Affiliations:** 1Institute of Biochemistry and Genetics UFRC RAS, Ufa 450054, Russia; koryakov_igor@mail.ru (I.K.); yuldashevra@gmail.com (R.Y.); iravtuschenko@mail.ru (I.A.); albinayakupovaa@yandex.ru (A.Y.); 2Department of Biology, Ufa University of Sciences and Technology, 32 Zaki Validi, Ufa 450076, Russia

**Keywords:** endophyte, *Bacillus subtilis*, cadmium acetate, *Triticum aestivum* L., growth, tolerance, oxidative stress, lignin, Cd accumulation

## Abstract

Heavy metal ions, in particular cadmium (Cd), have a negative impact on the growth and productivity of major crops, including wheat. The use of environmentally friendly approaches, in particular, bacteria that have a growth-stimulating and protective effect, can increase the resistance of plants. The effects of the pre-sowing seed treatment with the plant growth-promoting endophyte *Bacillus subtilis* 10-4 (BS) on cadmium acetate (Cd)-stressed *Triticum aestivum* L. (wheat) growth, photosynthetic pigments, oxidative stress parameters, roots’ lignin content, and Cd ions accumulation in plants were analyzed. The results showed that the tested Cd-tolerant BS improved the ability of wheat seeds to germinate in the presence of different Cd concentrations (0, 0.1, 0.5, and 1 mM). In addition, the bacterial treatment significantly decreased the damaging effects of Cd stress (1 mM) on seedlings’ linear dimensions (lengths of roots and shoots), biomass, as well as on the integrity and permeability of the cell walls (i.e., lipid peroxidation and electrolyte leakage) and resulted in reduced H_2_O_2_ generation. The pretreatment with BS prevented the Cd-induced degradation of the leaf photosynthetic pigments chlorophyll (Chl) a, Chl b, and carotenoids. Moreover, the bacterial treatment intensified the lignin deposition in the roots under normal and, especially, Cd stress conditions, thereby enhancing the barrier properties of the cell wall. This manifested in a reduced Cd ions accumulation in the roots and in the restriction of its translocation to the aboveground parts (shoots) of the bacterized plants under Cd stress in comparison with non-bacterized controls. Thus, the pre-sowing seed treatment with the endophyte BS may serve as an eco-friendly approach to improve wheat production in Cd-contaminated areas.

## 1. Introduction

Due to the increasing industrialization coupled with agricultural practices, soil contamination with heavy metals (HMs) has become a major environmental problem on a global level [1,2]. Throughout the world, there are 5 million sites of soil contaminated by heavy metals/metalloids with current concentrations above the regulatory levels [3]. One of the most toxic HM is cadmium (Cd), which is not an essential element for normal plant life and is not involved in plants physiological functions [1]. Plants’ common responses to Cd stress include abscisic acid accumulation, stomatal closure, braking of water absorption and transport, inhibition of the synthesis of chlorophyll and of the photosynthesis process, imbalance of pro- and antioxidants, and damage to the integrity of membrane structures, which leads to retardation of growth/development and reduction in plant yield/productivity [1,3,4,5,6,7,8]. Moreover, Cd ions may accumulate in all organs of a plant and thereby have a long-term negative effect on plant growth and yield quality [1,6,7,8,9,10,11,12]. The concentration of Cd in the soil is correlated with crop yield in many regions. For example, in the province of Henan (China), the content of Cd was 2.06 mg kg^−^^1^ of soil [4], while in Canada it was much lower (about 0.1 kg^−^^1^ soil) [7], and was correlated with the yields of major agricultural crops, including wheat, in these countries. Cereal crop cultivars around the world can store high Cd concentrations in grains [5,6,7,8]. More than 40% of Cd may be absorbed and transported to the upper parts of a plant and thus may directly (grains) or indirectly (animals) affect human health [6].

Wheat (*Triticum aestivum* L.) is the third most vital cereal in the world after rice and maize [4]. Almost 60% of the wheat produced globally is consumed as food [5,6], and the wheat demand is globally expected to rise by an estimated 70% in the next few decades, as the human population increases [6]. The accumulation of Cd ions in plants and grains of wheat has a negative impact on the health of people and animals that consume wheat [6,8,9]. The reduction of Cd in wheat is one of the main problems of sustainable agriculture and human health. During the last decades, various mitigation strategies such as the selection of wheat varieties with low Cd accumulation, the exogenous application of plant growth regulators (PGRs), the use of inorganic and organic additives, nanoparticles, and microbial agents (i.e., beneficial bacteria or fungi) have been applied to control Cd toxicity in wheat [6,9,13,14].

A beneficial role of plant growth-promoting microbes (PGPM) (endophytic and rhizospheric) in the reduction of HMs (including Cd) toxicity along with phytostimulation has been reported for several plants [14,15,16,17,18,19,20]. Numerous studies on the effects of microbes (bacteria or fungi) on plant growth under excessive HM action have shown their protective properties via the production of phytohormones, siderophores, 1-aminocyclopropane-1-carboxylic acid (ACC) deaminase, exopolymers, organic acids, biosurfactants, solubilizing phosphorus, and the modulation of plants’ antioxidant enzymes and HM resistance genes [20,21,22,23,24,25,26,27]. The potential mechanisms of plant survival in HM-contaminated soils may include two main pathways: the prevention of HM uptake (“avoidance”) and tolerance [28,29]. “Avoidance” restricts the translocation of metals into the above-ground parts of plants, protects the leaf tissue and, especially, photosynthetic cells from damage [28,29] This process involves reducing the absorption of HM ions by root cells by “capturing” them in the apoplast through the binding to organoacids or ion groups on the plant cell walls [30]. In contrast to limiting the input of toxic metals to plants, survival through tolerance involves physiological processes that allow a plant to function in the presence of high concentrations of HMs [26,29,31,32]. Tolerant plants are able to accumulate large amounts of HMs in the above-ground and underground masses through the synthesis of metal-binding compounds, their compartmentalization in less sensitive organs, and the production of antioxidants that counteract oxidative stress [29,33]. PGPMs may also use different mechanisms for establishing HM tolerance, e.g., exclusion, active removal, biosorption, and deposition or bioaccumulation of metals, in external and intracellular spaces [26,32,34,35,36,37]. These processes may affect the solubility and bioavailability of HMs to the plant, thereby altering their toxic effect [6]. For example, inoculation of wheat with *Ralstonia eutropha* and *Exiguobacterium aurantiacum* improved the plant growth and reduced the Cd ion input into the roots and shoots [16,35]. An important role in protecting plants from the toxic effects of HMs is played by the strengthening of their barrier properties through increased lignification [36]. However, the information on the influence of PGPMs on this process of plants under stresses is limited. As for PGPMs-inoculated wheat plants under Cd stress, data are still absent in the available literature. Some PGPMs can function as remediators by influencing the bioavailability of toxicants through the synthesis of chelates, siderophores, ACC deaminase, and indole-3-acetic acid (IAA) [15,25]. Elevated levels of toxic Cd and Ni were observed in *Brassica juncea* and *B. napus*, respectively, when the plants were inoculated with *Bacillus* spp. [38,39]. Some authors reported increased plant growth due to reduced accumulation of Cd in the shoots and roots of tomato plants inoculated with a *Burkholderia* sp. [40], but it is not clear how PGPM inoculation influences Cd accumulation in bacterized wheat plants. The findings indicate that the effects of PGPMs on HMs exposure, as well as the xenobiotic protection mechanisms in contaminated soils, may depend on both the bacterial species and the plant type. Additionally, microbes found in HM-contaminated soils may themselves be tolerant to such toxicants to a certain extent. HM-tolerant PGPMs are of practical importance both for the stimulation of plant growth and for the recovery of HM-contaminated soils [16,24,26,41]. Thus, despite studies on the general ways of increasing the tolerance of plants to HM stress, the specific response of a plant to the inoculation with specific species, as well as the microbe types and the stress factors involved, are of great interest and require further research. By today, the underlying mechanisms of the interaction between PGPMs and wheat plants under Cd stress are still not fully clear. Laboratory evaluations of the efficiency of HM accumulation may allow to choose the best associations of plant species with PGPMs for HM phytoremediation and plant protection [42,43].

Previously, we found that the pre-sowing seed treatment with the PGPM *B. subtilis* strain 10-4 had a growth-stimulating and anti-stress effect on various plants, including wheat under the influence of salinity [44,45], drought [46], and a combination of herbicide and drought [47]. It was shown that the protective effect of strain 10-4 is based on its ability to colonize the internal tissues of the host plant (including wheat), produce auxins, fix atmospheric nitrogen, solubilize phosphates, modulate the level of plant phytohormones (in particular, salicylic acid) [48] and the content of photosynthetic pigments, regulate the components of the ascorbate–glutathione complex [49], reduce stress-induced oxidative and osmotic cell damages, and also accelerate the lignification of root cell walls, as demonstrated for bean plants during salinity [50]. Based on the above-mentioned findings, we suggested that this bacterium has the potential to protect wheat plants against Cd toxicity as well. This work is both a continuation of the previous study on the mechanisms involved in the implementation of the effects of endophytic strain 10-4 and a new investigation of the ability of PGPMs to modulate growth and protect wheat plants under Cd stress.

This study aimed to analyze the effect of endophytic PGPM *B. subtilis* 10-4 (BS) on wheat growth, photosynthetic pigments, oxidative stress parameters, roots’ lignin content, and the accumulation of Cd ions in plants under Cd stress. 

## 2. Materials and Methods

### 2.1. Plant Material and Bacterial Strain

This work was carried out on hydroponically grown wheat (*Triticum aestivum* L., “Ekada70”) seedlings under laboratory conditions. The seeds were supplied by the Chishminsky Breeding Station UFRC RAS (Chishmy, Ufa, Russia). The endophytic bacterium *B. subtilis* 10-4 (BS) was previously isolated from dryland arable soils of the Republic of Bashkortostan (52°36′ N 58°19′ E, Russia) using a classical microbiological method [50], identified using 16S rRNA [44], characterized in detail [44,45], and deposited in the National Bio-Resource Center of the All-Russian Collection of Industrial Microorganisms (VKPM) with registration number B-12988. BS is able to colonize inner wheat tissues (endophyte) and induce plant growth-promoting traits, particularly, the production of auxins, siderophores, catalase and the fixation of atmospheric nitrogen [44].

### 2.2. Inoculum Preparation and Seed Treatment

Cells of BS were incubated in liquid Luria–Bertani (LB) medium for 24 h (37 °C, 180 rpm) until the cell concentration reached 10^9^ colony-forming units (CFU mL^−1^). Thereafter, the bacterial suspension was diluted (using sterile dH_2_O) to 10^5^ CFU mL^−1^ (previously selected as the optimal level for plant growth promotion and protection) [44]. The concentration of the bacterial cells was determined at OD 600 nm (SmartSpecTM Plus spectrophotometer, Bio-Rad, Hercules, CA, USA).

The wheat seeds were sterilized in 96% ethyl alcohol for 1 min, then washed with dH_2_O for 4–5 times. Thereafter, the seeds were immersed into solutions of BS (10^5^ CFU mL^−1^) or in dH_2_O (control) for 1 h.

### 2.3. Cd Stress Tolerance: Bacterial Strain Growth and Seed Germination In Vitro

The Cd tolerance of BS was assessed in triplicate by observing its growth in liquid LB medium containing different concentrations of Cd (0, 0.1, 0.5, and 1 mM). In sterilized flasks containing liquid LB medium (100 mL) with different Cd concentrations, we added 10 µL of a freshly cultured bacterial suspension (10^9^ CFU mL^−1^) and incubated the suspension at 37 °C and 180 rpm during 24 h. After 1, 3, 5, and 24 h of incubation, the optical density of the cells was measured at 600 nm (SmartSpecTM Plus spectrophotometer, Bio-Rad, USA). Additionally, the ability of the bacteria to grow in Petry dishes with solid LB medium containing different Cd concentrations (0, 0.1, 0.5, and 1 mM) was visually observed and photo-documented (PowerShot SX540 HS Digital Camera, Canon, Huntington, NY, USA). The significant growth of the bacteria in the presence of Cd during 24 h at 37 °C was considered as a demonstration of Cd tolerance.

The Cd tolerance of seeds during germination was measured by the ability of the seeds to germinate in solutions of Cd (0, 0.1, 0.5, and 1 mM). Bacterium-treated and -non-treated seeds were sown in Petri dishes with 5 mL of Cd solutions (tests) and water (control) (15 seeds per dish, three replicates). The seeds were grown for 3 days in the dark at 22 °C, after which the number of germinated seeds was counted. The percentage of germination was determined by the number of seeds producing a rootlet of the smallest length [51].

### 2.4. Design of the Experiments and Growth Conditions

The bacterium-treated seeds were hydroponically grown (on filter paper moistened with water) at 22–24 °C for 4 d (16 h light/8 h dark; 200 μmoL m^−2^ s^−1^). Thereafter, the seedlings were transferred to glasses with dH_2_O (control) or 1 mM Cd acetate [Cd(CH_3_COO)_2_] (Cd stress) and grown further in the same conditions.

Plant samples (i.e., roots, shoots (leaves), or whole seedlings) were taken after 24 h of Cd stress exposure to assess physio-biochemical attributes.

### 2.5. Growth Parameters

The length of the seedlings (roots and shoots) and their fresh (FW) and dry (DW) biomass were assessed by classical methods [51]. Each variant included 30 seedlings in three biological replicates.

### 2.6. Leaves Photosynthetic Pigments

The leaves (0.05 g) of wheat seedlings were homogenized in 90% ethanol (10 mL) with the addition of CaCO_3_ and filtered. The optical density of the filtered extracts was measured using a SmartSpecTM Plus spectrophotometer (Bio–Rad, USA) at 663 nm (chlorophyll a (Chl a)), 646 nm (chlorophyll b (Chl b)), and 470 nm (carotenoids (Car)). The concentration of the pigments was expressed as mg g^−^^1^ FW [52,53].

### 2.7. Lignin Content and Deposition in the Cell Wall of the Basal Part of the Roots

The qualitative analysis of lignin deposition in the cell walls of the basal roots of 4–5-day-old wheat seedlings was based on the intensity of the red-purple coloration of lignin by floroglucinol using a light microscope (Amprival Carl Zeiss, Jena, Germany) [54]. The level of lignin deposition was assessed on a color scale [55].

The qualitative analysis of lignin content in the roots was performed by following the method by Sancho et al. [56]. About 100 mg of root tissues was thoroughly rinsed in hot water and, after centrifugation, the insoluble particles were pelleted and rinsed in 100% ethanol. The dry residue thus obtained was solubilized for 2.5 h in a solution of 2.5 mL of HCl/ethanol. Then, 10 μL of 20% phloroglucinol–HCl was mixed with 1 mL of the previous solution. After 30 min of incubation, the absorbance of the mixture was recorded at 540 nm (UV-2800 Unico spectrophotometer, United Products & Instruments, Dayton, NJ, USA).

### 2.8. Lipid Peroxidation (MDA)

The lipid peroxidation degree was determined by the content of malondialdehyde (MDA) [57]. Whole wheat seedlings were ground in dH_2_O and homogenized in 20% trichloroacetic acid. The homogenized samples were then centrifuged (10,000× *g*, 10 min). The supernatant was mixed with 0.5% thiobarbituric acid prepared in 20% trichloroacetic acid and kept in a boiling water bath (100 °C, 30 min); then, it was quickly cooled. Absorbance was measured at 532 nm and 600 nm (SmartSpecTM Plus spectrophotometer, Bio-Rad, CA, USA). The concentration of MDA was calculated using an extinction coefficient of 155 mM^−^^1^ cm^−^^1^ and is expressed as nmoL g^−^^1^ FW.

### 2.9. Hydrogen Peroxide (H_2_O_2_)

H_2_O_2_ was determined according to [58]. The roots of wheat seedlings were homogenized (1:5 *w*/*v*) in 0.05 M sodium phosphate buffer (pH 6.2). The supernatant was separated by centrifugation (Eppendorf^®^ Microcentrifuge 5415R, Humburg, Germany) at 15,000× *g* for 15 min. The concentration of H_2_O_2_ in the supernatant was measured at 560 nm (SmartSpecTM Plus spectrophotometer, Bio-Rad, Hercules, CA, USA) using xylenol orange in the presence of Fe^2+^. H_2_O_2_ is expressed as µmoL g^–1^ FW.

### 2.10. Electrolytes Leakage (EL)

The disturbance of the barrier properties of the cell membranes was evaluated by following EL from plant tissues, recorded using the HI8733 conductivity meter (Hanna Instruments Inc., Szeged, Hungary) [59]. Whole seedlings of wheat (1 g) were washed with running H_2_O and cut into equal-size fragments, then washed again with running H_2_O for 3 min, rinsed with dH_2_O, slightly dried, supplied with 20 mL of dH_2_O, and incubated at 25 °C for 1 h. Then, the samples were filtered, and the electroconductivity of the obtained solutions was measured and expressed as μSi g^−1^ FW.

### 2.11. Cd Content in Plants Tissues

The Cd content was determined by atomic absorption spectrophotometry [54] in the roots and shoots of wheat seedlings pretreated and untreated with BS and exposed to 1 mM Cd stress during 24 h. Air-dried plant samples (1 g) (roots or shoots) were placed into a muffle furnace for 20 h at 550 °C. Thereafter, the obtained ash was incubated with 0.1 M HNO_3_ (50 mL) and filtered. The concentration of Cd was determined in the filtered extracts using an atomic absorption spectrophotometer (Shimadzu AA-6300, Shimadzu, Kyoto, Japan). The content of Cd is expressed as mg g^−^^1^ DW. 

### 2.12. Statistical Analysis

The data were analyzed by a one-way ANOVA test using the SPSS 21.0 software. All statistical differences are presented with respect to the untreated control. The means were recorded for three replicate values and were compared using Duncan’s multiple range test; statistically significant differences were evaluated at (*p* ≤ 0.05). The calculation of the Pearson’s correlation coefficients and the construction of correlation matrices were carried out using Microsoft Excel STATISTICA 10.0 software.

## 3. Results

### 3.1. Cd Tolerance Test: Ability of Bacterial Cells and Wheat Seeds to Grow in the Presence of Different Concentrations of Cd Acetate

The evaluation of tolerance to Cd showed that endophytic BS strain cells were capable to grow in the presence of different concentrations of Cd both on liquid (Figure 1A) and on solid (Figure 1B) Luria–Bertani (LB) medium. There was a concentration-dependent decline in bacterial cells’ growth over the cultivation period (24 h) in liquid LB medium (Figure 1A), but the BS strain remained able to grow in the presence of Cd (although more slowly than the controls) demonstrating the slowest growth in the presence of 1 mM Cd. The assessment of BS growth on solid LB medium with Cd acetate after 24 h of incubation showed the same results (Figure 1B).

Further tests showed that wheat seed germination was significantly affected by various concentrations of Cd stress as well (Figure 1C). With the increase in the concentration of Cd (from 0 to 1 mM), the percentage of seed germination gradually decreased. However, the pretreatment with BS significantly improved seed germination in the presence of all tested concentrations of Cd acetate (0. 0.1, 0.5, and 1 mM).

Thus, the results demonstrated that the bacterial endophyte was Cd-tolerant and might protect wheat seed germination in Cd-contaminated conditions upon the pre-sowing seed treatment.

### 3.2. The Endophyte B. subtilis Improved Wheat Growth under Normal and Cd Acetate Stress Conditions

Further analysis carried out in hydroponically growth wheat seedlings that were pretreated with BS, grown under normal conditions for 4 days, and subjected to 1 mM Cd acetate stress for 24 h. The results demonstrated that under normal growth conditions, pretreatment with the endophyte led to increased length of the roots (by 1.5 times), shoots (by 1.3 times), and whole seedlings (by 1.4 times) in comparison with the control (Figure 2A). This was confirmed by a visual evaluation of the seedlings (Figure 2B). Similar results were obtained in the estimation of their fresh (Figure 2C) and dry (Figure 2D) biomass. 

Exposure to Cd stress had a pronounced toxic effect on the wheat seedlings and inhibited their growth (Figure 3). The roots’, shoots’, and whole seedings’ length (Figure 3A,B) and their fresh (Figure 3C) and dry biomass (Figure 3B) were about 1.2–1.3 times lower the control values. A comparison of the growth rates of BS-pretreated plants not exposed to and exposed to Cd stress revealed that the pretreatment with BS did not prevent, but significantly reduced, the negative action of Cd stress on the seedlings and helped to maintain the intensity of the growth processes of these plants at least at the level of those in the control (Figure 3). Particularly, the length (Figure 3A,B) and biomass (Figure 3C,D) of roots, shoots, and seedlings exposed to the same Cd stress upon BS pretreatment were about 1.2 times higher than the control values. Thus, the BS pretreatment had a clear protective effect on wheat growth under conditions of Cd stress.

### 3.3. Effect of Cd Stress on Leaf Photosynthetic Pigments in B. subtilis-Treated Wheat

Cd stress adversely affected the leaf pigment system in 5-day-old wheat seedlings exposed to Cd stress for 24 h (Figure 4). A significant decline in chlorophyll a (Chl a) (by 2.5 times) (Figure 4A), chlorophyll b (Chl b) (by 1.6 times) (Figure 4B), carotenoids (Car) (by 1.3 times) (Figure 4C), and the total amount of pigments (TAP) (by 2 times) (Figure 4D) was observed in metal-stressed seedlings.

The treatment with BS decreased the level of stress-caused depletion of all studied pigments in the wheat leaves. The contents of Chl a and Chl b in the leaves of the bacterium-treated and stressed plants were about 15% and 75% below the control levels, respectively, while, the content of Car in BS-pretreated and stressed plants was maintained at the control level (non-stressed plants). Under normal growth conditions, there was a slight increase in the content of the studied pigments upon BS treatment (Figure 4).

### 3.4. Effect of the Endophyte B. subtilis on Lignin Deposition and Content in Wheat Roots under Cd Stress

Since the BS treatment reduced the degree of the damaging action of Cd on the growth processes of wheat seedlings, it was hypothesized that the protective effect of BS on plants under Cd stress could be associated with an increase in the barrier properties of root cell walls. It is shown (Table 1) that the introduction of Cd acetate on the incubation medium of 4-day-old seedlings for 24 h (1 day) resulted in the acceleration of lignin formation, which was first observed in the cell walls of the central cylinder. Pretreatment with BS also contributed to the acceleration of lignin deposition in the cell walls of the central cylinder of the roots of 4–5-day-old seedlings under normal growth conditions. At the same time, 5-day-old BS-pretreated seedlings exposed to 1 mM Cd acetate were characterized by an additional intensification of lignin deposition in the cells of the central cylinder and in the nearby cells of the pericycle. The content of lignin in the wheat roots of all studied variants changed in a similar manner (Table 1).

### 3.5. Levels of Oxidative Stress Markers in the Presence and Absence of the Endophyte B. subtilis under Cd Stress

Cd stress led to more-than-two-times higher accumulations of H_2_O_2_ (Figure 5) and MDA (Figure 6A) and to an increase in the rate of electrolyte leakage (Figure 6B). Pretreatment with the endophyte BS reduced the Cd stress-induced accumulations of H_2_O_2_ (by 1.4 times) and MDA (by 1.3 times). The membrane permeability in these plants was also decreased (by 1.4 times) in comparison with control (non-bacterized) stressed plants (Figure 6). Under normal (non-stressed) growth condition, the bacterial inoculation did not affect the level of the oxidative stress markers H_2_O_2_ (Figure 5), MDA (Figure 6A), and electrolyte leakage (Figure 6B).

### 3.6. Effect of the Endophyte B. subtilis on Cd Uptake by the Wheat Seedlings

To determine the effects of the endophyte BS inoculation on the accumulation of Cd ions in wheat seedlings, the total metal concentrations in the roots and shoots were determined. The data presented in Figure 7 demonstrate that the maximal accumulation of toxic Cd ions was observed in the wheat roots, while in the shoots, the content of Cd was significantly (by 10 times) lower than in the roots. At the same time, the pretreatment with BS significantly reduced the Cd concentration both in the roots (by 2 times) and in the shoots (by 3.1 times) in comparison with the non-treated and Cd-exposed controls (Figure 7).

### 3.7. Correlation Matrices

A factor analysis showed that all studied indicators (Figure 8) could be divided into two clusters, with a mutual positive correlation within them and a negative correlation between them. The first cluster included growth indicators, such as length, FW, and DW of shoots and roots, as well as the content of the photosynthetic pigments Chl a and Car. The second cluster included the content of Chl b, of lignin in the roots, of Cd in the shoots and roots, as well as the levels of markers of oxidative stress, i.e., H_2_O_2_, MDA, and EL. In the first cluster, quite expectedly, the content of the photosynthetic pigments Chl a and Car correlated more with the growth rates of the shoots than with that of the with roots. As for the second cluster, attention was drawn to the fact that the lignin content correlated to the least extent with other indicators within the cluster. This can be explained by the fact that this indicator is not strictly stress-induced, since also the treatment with the bacteria caused (intensified) the deposition of lignin in the roots (Table 1). 

## 4. Discussion

The physiological activity of bacteria is an important indicator to evaluate the feasibility of using them to stimulate the host plant growth and its tolerance to possible stress factors. Bacteria that have a protective effect on plants under HM stresses also have a certain tolerance to the action of these pollutants [60]. Microorganisms exhibit tolerance to HMs through their immobilization on the cell surface or transformation into less toxic forms, for example, by binding them (precipitation) or using reactions of oxidation–reduction [60,61]. For example, ACC deaminase-producing *Pseudomonas* spp. are tolerant to Cr, Cd, Cu, and Pb toxicity and are able to increase the growth of wheat [62]. In a different study, the growth-stimulating capacity of PGPMs in wheat plants under Cd stress significantly enhanced root elongation and minimized the ethylene level in seedlings [63]. We observed a reduction in the growth of the tested bacterium BS with increasing Cd concentrations, with the highest inhibiting effect in the presence of Cd 1 mM (Figure 1), which is consistent with literature data about the influence of HMs on *Bacillus* microorganisms [17,37]. However, the endophyte that colonized the plant tissues after using the pre-sowing seed treatment technique, as showed earlier [63], was able to withstand the effect of 1 mM Cd and exerted a prolonged protective effect on the wheat seedlings (Figure 3). The growth-promoting effect of endophytic BS on plants is probably associated with the ability of this strain to produce various biologically active substances [34] such as IAA and siderophores, as well as to fix atmospheric N [44], which modulates plant responses and is responsible for establishing interactions between plants and microbes [34]. *Bacillus* spp. may use the produced auxins to interact with plants as part of their colonization strategy, including phytostimulation and bypassing major plant defense mechanisms [34]. For example, Ikram et al. (2018) reported that the IAA-producing endophytic fungus *Penicillium ruqueforti* induced great resistance in wheat plants grown in HM- (Ni, Cd, Cu, Zn, and Pb) contaminated soil, improving plant growth and also nutrient uptake. Moreover, *P. ruqueforti* restricted the transfer of HMs (Ni, Cd, Cu, Zn, and Pb) from the soil to the plants (shoot and roots) by secreting IAA [23].

Photosynthesis is a fundamental process that determines plant growth [64]. The decay in chlorophyll content and subsequent inhibition of photosynthesis are symptoms of HMs toxicity and typically reduce plant growth [4]. It was reported that the photosynthetic pigments chlorophyll a, b, and total chlorophyll were drastically reduced in wheat flag leaf with increasing the concentrations of externally supplied metals, and the least contents were measured in the presence of Pb, followed by Ni and Cd [11]. This decrease can be explained by (i) the disturbance of the absorption of Mg2+ and essential elements (such as Fe and Mn), (ii) the inhibition of chlorophyll biosynthesis as a result of HMs interacting with the –SH group of enzymes involved in this process, and (iii) the increased activity of chlorophyllase (the first enzyme in the catabolic route of chlorophyll), which led to the degradation of chlorophyll. In our study, under conditions of Cd stress, the pretreatment with BS contributed to a maintaining the level of pigments above the stress level (Figure 4). The endophyte *Glomus mosseae* improved wheat growth parameters, increased chlorophyll content, and reduced Cd content in shoots under Cd toxicity [21]. Inoculation with endophytic *Talaromyces pinophilus* boosted wheat plant growth features and the levels of photosynthetic pigments, osmolytes (soluble proteins, soluble sugars, and total amino acids), enzymatic antioxidants (catalase, superoxide dismutase, and peroxidase), K, Ca, and Mg [54]. On the other hand, it reduced Na, the Na/K ratio, Cd, Ni, Cu, and Zn in the growth medium as well as in the shoots and roots of wheat. These results suggest that endophytic *T. pinophilus* can work as a barrier to reduce the absorption of HMs in wheat cultivated in soil amended with sewage sludge [65].

Along with a decrease in photosynthetic activity, Cd ions caused the development of oxidative stress in wheat, as evidenced by the data presented in Figure 5 and Figure 6. This is consistent with other reports that showed that exposure to toxic Cd ions resulted in the over-production of reactive oxygen species (ROS) and enhanced lipid peroxidation, the intensity of which can be judged by the level of MDA as one of the main end products of lipid peroxidation (LPO) [10,66]. The MDA levels in plants treated with the bacterium, grown both in the absence and in the presence of Cd stress, were lower than in the untreated control, i.e., the bacterization of the seeds contributed to maintaining the integrity of the membrane structures of the plant cells. This was probably due to the ability of BS during pretreatment to cause a balanced increase in O^2−^ and H_2_O_2_ generation and to induce the activity of antioxidant enzymes in the wheat seedlings, involved in the neutralization of the oxidative explosion caused by Cd [32]. The local generation of a low level of H_2_O_2_ in this case, apparently, not only improved the plasmodesm transport capacity but was also used by peroxidase to modify the cellular wall, pre-adapting it to subsequent stress [67]. The endophytic *B. subtilis* 11VM and 26D improved *Sinapsis alba* tolerance to Cd and Ni toxicity and reduced oxidative stress in the presence of high levels of metal ions in the aboveground parts of plants [68]. Other authors showed that a *B. subtilis* inoculation of *Oryza sativa* L. could highly reduce Cd accumulation in every part of the plant roots and shoots (45 days) and in grains (120 days) [69]. *B. subtilis* can effectively absorb Cd compared to *B. cereus*, which might be the main mechanism to reduce Cd transportation in rice plants. Moreover, *B. subtilis* could increase rice plant biomass and protect from Cd stress due to its ability to produce IAA, solubilize phosphate, and control the ethylene levels by ACC deaminase activity [69].

Since the BS treatment reduced the degree of the damaging action of Cd on the growth processes of the wheat seedlings, it was suggested that the protective effect of BS on plants under Cd stress is associated with an increase in the barrier properties of the root cell walls. An important role in protecting plants from the toxic effects of HMs is played by the strengthening of their barrier properties through increased lignification [36]; however, information on the influence of endophytes on this process is limited. There are few data on the effect of PGPM on the lignification of plant cells. For example, the rhizobacteria *Pseudomonas aeruginosa* and *B. megaterium* protected maize plants from salt stress-caused damages along with regulating cell water content, phenols, flavonoids, and antioxidant enzymes, also through lignification [70]. Inoculation with the plant growth-promoting rhizobacteria cytokinin-producing *B. subtilis* and auxin-producing *P. mandelii* strains resulted in a faster appearance of Casparian bands in the root endodermis and an increased growth of the salt-stressed plants [27]. Inoculation with the endophytic *B. subtilis* 10-4 increased bean plants’ tolerance to NaCl salinity due to the strengthening of the root cell walls through increased lignin deposition [44]. However, we had no data on the effect of endophytic PGPM on wheat plants under Cd stress conditions by the time we started our work. Wheat seed inoculation with BS accelerated cell wall lignification relative to control plants. We believe that this ability of this bacterium to strengthen the barrier properties is fundamentally important when plants are exposed to stress. Indeed, under conditions of Cd stress, additional accumulation of lignin was observed in plants pretreated with the bacterium. Lignification affects not only the cells of the central cylinder but also those of the pericycle (Table 1). This reaction of bacterium-inoculated plants to the presence of Cd ions is considered important for plant protection against HMs. This was also reflected in the levels of the markers of oxidative stress (Figure 5 and Figure 6). Thus, the obtained results showed the involvement of BS in the regulation of lignin formation, which obviously makes an important contribution to the protection by these bacteria of wheat plants from Cd-caused osmotic and oxidative cell damages, and in the reduction of Cd accumulation in the roots and its further translocation into the above-ground parts of the plants due to the fact that the roots have a strong barrier that prevents the penetration of toxic ions into the cells. These observed effects are certainly of interest, and further close attention and in-depth research in this direction are recommended for a more complete use of the potential of the endophytic bacterium *B. subtilis* in environmentally oriented technologies for growing wheat, especially under adverse stress factors, including Cd toxicity. 

Soil contamination with HMs inevitably leads to their accumulation in the soil and plants and further transfer through the food chain [71]. Numerous studies in recent years have proved that the bacteria used in agriculture can actively influence not only plant growth, but also the degree of absorption of metals [72]. Cd may easily be absorbed by plants via root uptake and be translocated to shoots and grains because of its high mobility [6]. Many studies have indicated that a high Cd content in wheat is reached through the roots, and Cd may translocate to the shoots and grains [6,73]. It was reported that metal(loid)-resistant bacteria (*Ralstonia eutropha* and *Exiguobacterium aurantiacum*) decreased the Cd uptake by wheat and diminished Cd availability in soils and its accumulation in plants [16,35]. It was noted that metal-tolerant bacteria can interact directly with HMs to diminish their toxicity or modulate their bioavailability [15,16]. Researchers [15,16] found that metal(loid)-resistant bacteria have a vital role in increasing plant biomass and toxic metal tolerance by producing siderophores, ACC deaminase, and IAA. References [13] and [41] reported that *Bradyrhizobium*, *Rhizobium*, *Pseudomonas*, and *Stenotrophomonas* can be used to reduce metal accumulation in plants. Other studies [24] showed that the endophytic bacteria *Exiguobacterium indicum* SA22 and *Enterobacter ludwigii* SAK5 are effective in the reduction of Cd accumulation in rice. Some authors showed that plant-associated bacteria increased the growth of the host organism (plant), but the effect of metal absorption depended on the specific plant–microbial partnership, as well as on the physiochemical properties of the soil [74]. The bioavailability of a metal is often specific and is determined by the type of plant, the element (the metal itself), and the activity of bacteria capable of synthesizing biologically active substances [75]. In our study, the inoculation of wheat seeds with endophyte BS reduced the supply of Cd ions to the roots and shoots of the wheat plants (Figure 7). It can be assumed that the ability of endophytes to accelerate lignification, discovered during this study (Table 1), leads to an increase in the barrier properties of the plant cell walls, which leads to a much lower Cd accumulation in these plants, especially in the shoots (Figure 7). Under the influence of BS, apparently, an avoidance mechanism was triggered: due to lignification of the roots, the barrier properties were strengthened, and the entry of toxic ions into the roots and their further movement into the aerial parts of plants was limited. These was reflected in the prevention of the decrease in the photosynthetic pigments (Figure 4) and in the reduction of the degree of the damaging effect of Cd on the integrity and permeability of the membrane structures of the cells (Figure 5 and Figure 6) and on the growth of the wheat seedlings (Figure 3).

## 5. Conclusions

Our study revealed that BS exhibits tolerance to Cd ions. A seed pretreatment with BS had a protective effect on wheat plants against Cd stress. Obviously, the ability of BS to strengthen the barrier properties of wheat plant cell walls due to the intensification of lignin deposition in normal conditions played a key role under stress (Cd) conditions. The additional accumulation of lignin under stress in BS-treated plants led to a decrease in the accumulation of Cd in the roots and, especially, in the shoots, which made an important contribution to reducing the damaging effect of Cd on the photosynthetic apparatus and the state of membrane structures, as evidenced by the stabilization of the growth of these plants.

## Figures and Tables

**Figure 1 microorganisms-11-01653-f001:**
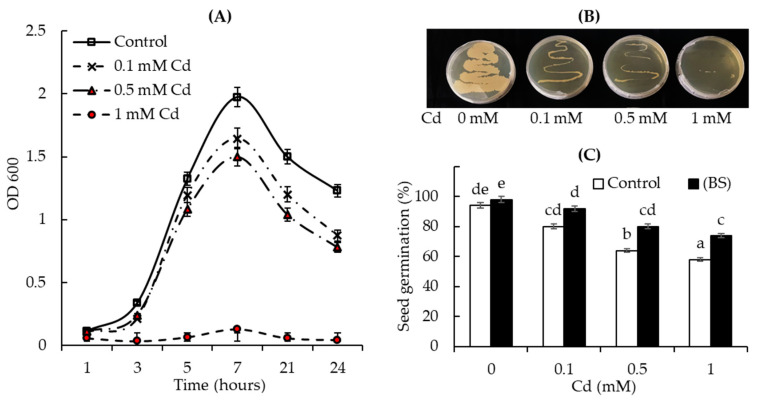
Growth of *B. subtilis* 10-4 (BS) cells in the presence of cadmium acetate (0, 0.1, 0.5, and 1 mM) in liquid LB medium (determined at OD 600 nm) (**A**) and photographs of bacterial growth in solid LB medium containing different Cd concentrations (0, 0.1, 0.5, and 1 mM) (after 24 h of incubation) (**B**); effect of the pretreatment with BS on the percentage of wheat seed germination in the presence of different Cd concentrations (0, 0.1, 0.5, and 1 mM) (**C**). The data were obtained in triplicate for each treatment. Error bars represent standard errors (±SE) of the means. Various letters under the columns indicate a significant difference between the averages of different groups at *p* < 0.05.

**Figure 2 microorganisms-11-01653-f002:**
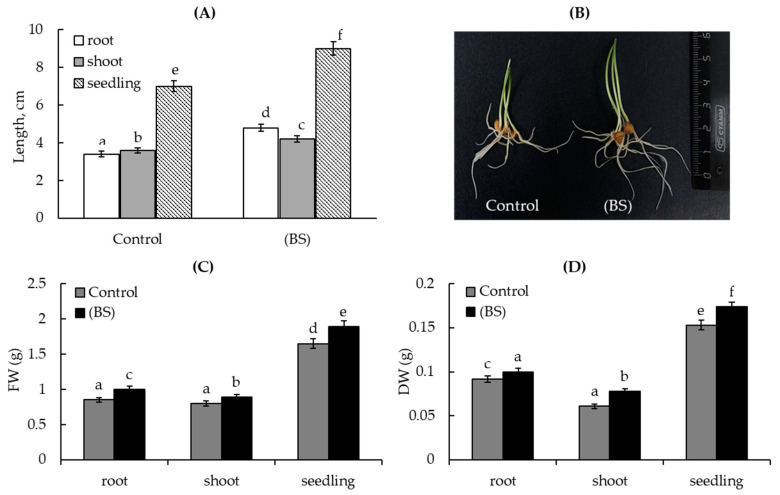
Effect of the pre-sowing seed treatment with the endophyte *B. subtilis* (BS) on the growth parameters of 4-day-old wheat seedlings under normal grow conditions: (**A**) length of the roots, shoots, and seedlings; (**B**) visual appearance of 4-day-old seeglings; (**C**) fresh biomass (FW) and (**D**) dry biomass (DW) of roots, shoots, and seedlings. In the picture the seedlings of three repetitions (n = 30) are presented. Control—control plants (without bacterial treatment). BS—plants pretreated with *B. subtilis*.

**Figure 3 microorganisms-11-01653-f003:**
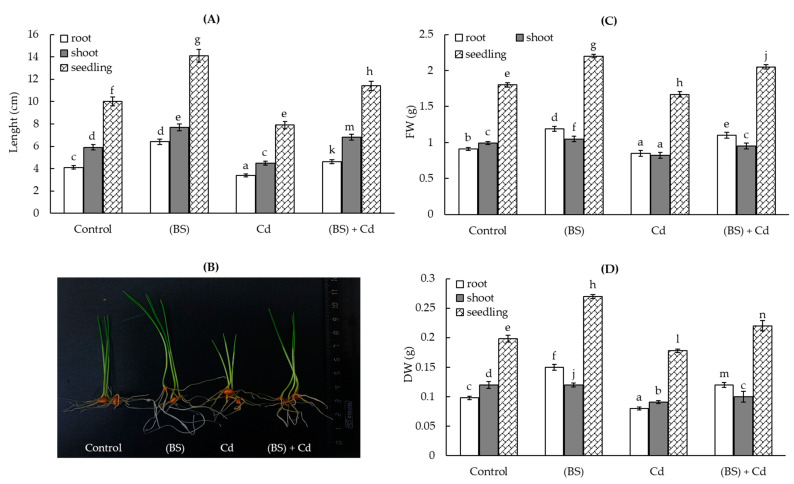
The changes in the length of the roots, shoots, and seedlings (**A**), their visual appearance (**B**), and fresh (**C**) and dry (**D**) biomass for 5-day-old wheat seedlings pretreated with *B. subtilis* (BS) and exposed to Cd stress for 24 h (i.e., 4-day-old seedlings grown under normal conditions were transferred into 1 mM Cd(CH_3_COO)_2_ for 24 h). All the statistical differences are presented relative to the untreated control. Columns of each histogram marked with different letters represent the mean values that were statistically different from each other according to the Duncan’s test (*p* ≤ 0.05). The picture shows the results for the seedlings of three repetitions (n = 30). Control—control seedlings (non-bactrerial treated).

**Figure 4 microorganisms-11-01653-f004:**
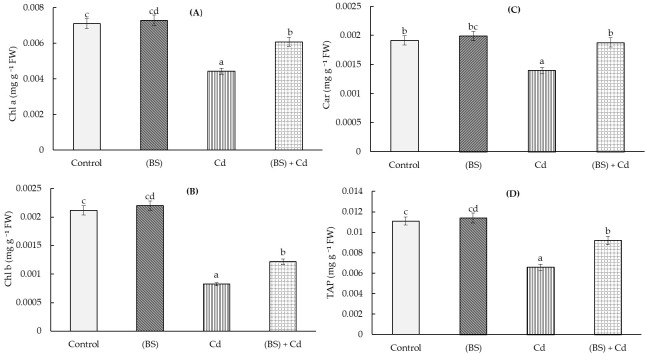
Influences of the seed treatment with the endophyte *B. subtilis* 10-4 (BS) on the content of the photosynthetic pigments chlorophyll (Chl) a (**A**), Chl b (**B**) and carotenoids (Car) (**C**) and on the total amount of pigments (TAP) (**D**) in leaves of 5-day-old wheat seedlings under normal conditions and 1 mM cadmium acetate (Cd) stress. Time of stress exposure—24 h. All the statistical differences are presented relative to the non-bacterium-treated control. Columns of each histogram marked with different letters indicate the mean values that were statistically different from each other according to the Duncan’s test (*p* ≤ 0.05).

**Figure 5 microorganisms-11-01653-f005:**
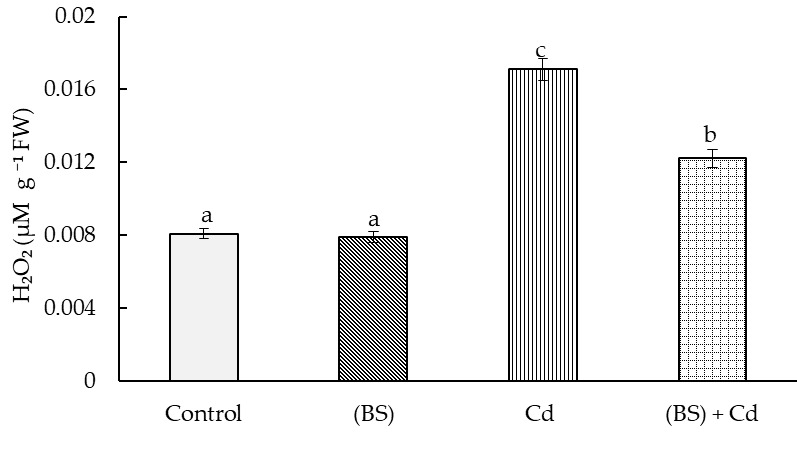
Influence of the endobacterium *B. subtilis* 10-4 (BS) pre-treatment on hydrogen peroxide (H_2_O_2_) in the roots of 5-day-old wheat seedlings under normal conditions and 1 mM cadmium acetate (Cd) stress. Time of stress exposure—24 h. All the statistical differences are presented relative to the non-bacterium-treated control. Columns of each histogram marked with different letters indicate the mean values that were statistically different from each other according to the Duncan’s test (*p* ≤ 0.05).

**Figure 6 microorganisms-11-01653-f006:**
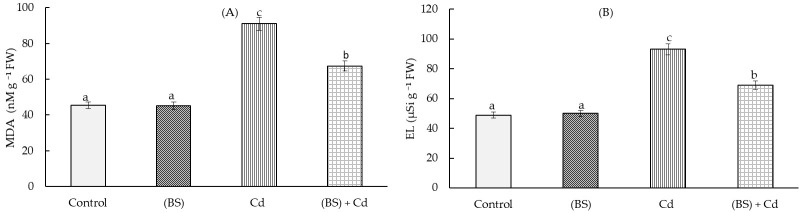
Effect of the endobacterium *B. subtilis* 10-4 (BS) pre-treatment on MDA content (**A**) and electrolyte leakage (**B**) in 5-day-old wheat seedlings under normal conditions and 1 mM cadmium acetate (Cd) stress. Time of stress exposure—24 h. All the statistical differences were presented relative to the non-bacterium-treated control. Columns of each histogram marked with different letters indicate the mean values that were statistically different from each other according to the Duncan’s test (*p* ≤ 0.05).

**Figure 7 microorganisms-11-01653-f007:**
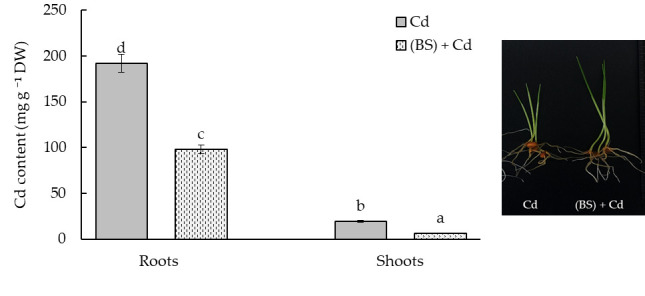
Cadmium (Cd) concentration in the roots and shoots of 5-day-old wheat seedlings untreated and pretreated with the endophyte *B. subtilis* 10-4 (BS) and exposed to 1 mM Cd acetate for 24 h. All the statistical differences are presented relative to the untreated control. Columns of each histogram marked with different letters indicate the mean values that were statistically different from each other according to the Duncan’s test (*p* ≤ 0.05).

**Figure 8 microorganisms-11-01653-f008:**
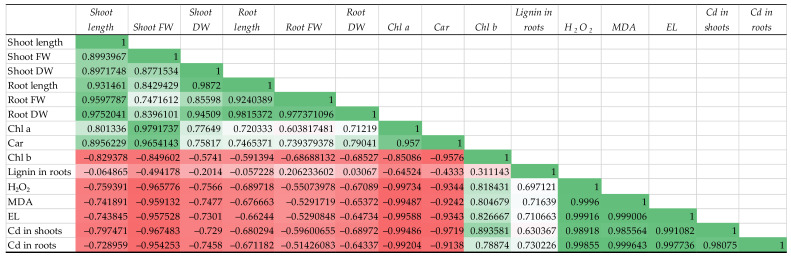
Correlation matrix representing the correlation between key parameters examined in this study.

**Table 1 microorganisms-11-01653-t001:** Effect of *B. subtilis* 10-4 (BS) on the lignin content in the roots under normal and Cd stress conditions. Lignin deposition was determined in the cell walls of the basal part of the roots of 4- and 5-day-old seedlings. Values are means ± SE (n = 10). The variants in the same column marked with different letters represent the mean values that are statistically different from each other according to the Duncan’s test (n = 10, *p* ≤ 0.05).

Treatment	Lignin Content (_Δ_A_540 nm/_g FW)		Visual Assessment of Lignin Deposition in Roots
	4-Day-Old	5-Day-Old	
Control	0 ^a^	0 ^a^	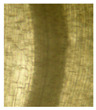
(BS)	1.12 ± 0.045 ^b^	1.21 ± 0.049 ^b^	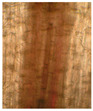
Cd	N/A	1.88 ± 0.079 ^c^	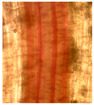
(BS) + Cd	N/A	2.1 ± 0.094 ^d^	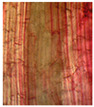

“N/A”—not accessed.

## Data Availability

Not applicable.

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
