# Peer review of "Endophytic Plant Growth-Promoting Bacterium Bacillus subtilis Reduces the Toxic Effect of Cadmium on Wheat Plants"

_microorganisms, 2023, doi:10.3390/microorganisms11071653_

Round 1
Reviewer 1 Report
Dear Auhtors,
The introduction part is prepared in a sufficient way and gives the reader clear background to results analysis; The aim of the studies is clearly presented.
In general manuscript is well written, but some aspects should be clarified or more detailed described in materials and methods and also in results part:
The material and methods sections is detailed and clearly presented; One suggestion is to precise add information- where the material is only roots tissues, where leaves, because in some methods Author used term ‘plants samples’, ‘plant tissues’ or ‘seedlings’- so we compared different parameters from different plants part ?
Results section:
Roots analyses is understandable in the aspect of stimulation of plant growth and recovery of HM-contaminated soils, but despite of it - It will be also very interesting how Cd and bacterial treatment have or not have influenced on above-ground parts of the plant, which the most important in wheat breeding, why Authors concentrated on lignin content only on roots ? Moreover, what about lack of lignin content in the control, isn’t it correct that in 5 or 4 old seedlings lignin was not present in cell walls?
How to explain that the differences between control plant and plant with endophyte are strongly limited?
Instead of only table content (table 1 and also 2) with statements ”weak”, “very weak”, “strong” staining based on spectrophotometer analysis it should be also added staining on tissues section especially since the Authors received phloroglucinol, it will not be complicated;
I warmly suggest to unified the results presentation and from hydrogen peroxide, MDA and EL also add charts presentation- it will be easier for the reader to catch the significance differences;
What kind of future prospects can be assumed coming from obtained results?
Sincerely
Manuscript is well written only some minor English correction are needed;
Author Response
Dear Reviewer, thank you for the attentive attitude to our article and for your comments/suggestions. We made corrections to the manuscript according to yours and other Reviewers’ comments.
The introduction part is prepared in a sufficient way and gives the reader clear background to results analysis; The aim of the studies is clearly presented.
In general manuscript is well written, but some aspects should be clarified or more detailed described in materials and methods and also in results part:
The material and methods sections is detailed and clearly presented; One suggestion is to precise add information- where the material is only roots tissues, where leaves, because in some methods Author used term ‘plants samples’, ‘plant tissues’ or ‘seedlings’- so we compared different parameters from different plants part ?
Response: Clarifications is added now.
Results section:
Roots analyses is understandable in the aspect of stimulation of plant growth and recovery of HM-contaminated soils, but despite of it - It will be also very interesting how Cd and bacterial treatment have or not have influenced on above-ground parts of the plant, which the most important in wheat breeding, why Authors concentrated on lignin content only on roots ? Moreover, what about lack of lignin content in the control, isn’t it correct that in 5 or 4 old seedlings lignin was not present in cell walls?
Response: According to the results, bacterial treatments also positively influenced above-ground parts as well, as indicated the data about the length and biomass of shoots under normal (Figure 2A-D) and Cd stress conditions (Figure 3A-D), the content of leaf photosynthetic pigments (Figure 4), and lower Cd accumulation in shoots (Figure 7). Since in natural growing conditions heavy metals are mainly present in the soils and the roots affected the first to stress, we assessed lignification in the roots.
As for lack of lignification in basal part of roots of 4-5 days old control seedlings, probably its due to seedlings being in early stage are in the process of growth and there is not lignin yet, and if it starts to synthesizing but it was not enough to be visualized and did not detect by the used methods (probably the methods used is not enough sensitive and we could not see the small amounts of lignin in 5 days controls even if it starts to synthesizing). While in bacterial treated wheat we observed the beginning of lignification, that is probably associated with the ability of bacteria to intensify growth processes and 5-days bacterized seedlings older than that of control (non-bacterized) 5 days seedlings. Previously also was demonstrated that under normal growth conditions the lignification in wheat roots began to appear in 6th day, while upon treatment with growth regulators the lignification was appeared in 4th or 5th day of growth (Bezrukova et al. Lectin involvement in the development of wheat tolerance to cadmium toxicity. Russian Journal of Plant Physiology. 2011. Т. 58. № 6. С. 1048-1054. https://link.springer.com/article/10.1134/S1021443711060021; Maslennikova D. et al. Components of the phenylpropanoid pathway in the implementation of the protective effect of sodium nitroprusside on wheat under saline conditions. Plants 2023, 12, 2123. https://doi.org/10.3390/plants12112123). It is also possible that the speed of lignification process may be dependent on plant genotype and its varietal characteristics.
How to explain that the differences between control plant and plant with endophyte are strongly limited?
Response: Thanks for the question. We have previously shown that endophytes Bacillus subtilis (strain 10-4) live well inside plant tissues and have a positive effect on wheat. This presence of bacteria should be physiologically comfortable for the plants, so the bacteria have a possibly not very strong but persistent growth-stimulating effect on wheat.
Instead of only table content (table 1 and also 2) with statements ”weak”, “very weak”, “strong” staining based on spectrophotometer analysis it should be also added staining on tissues section especially since the Authors received phloroglucinol, it will not be complicated;
Response: We added the photos showing staining tissues section of wheat roots to modified version of the manuscript.
I warmly suggest to unified the results presentation and from hydrogen peroxide, MDA and EL also add charts presentation- it will be easier for the reader to catch the significance differences;
Response: The charts are added to the modified manuscript.
What kind of future prospects can be assumed coming from obtained results?
Response: 1) Seed priming with Cd tolerant endophyte B. subtilis may be used as an eco-friendly approach in environmentally-oriented wheat production under Cd contaminated areas. Resulting in both plant growth improvement and also the reduced Cd accumulation in food.
2) The strengthening of the barrier properties of cell walls is a universal mechanism for protecting plants from the toxic effect of not only cadmium ions, but sodium chloride and other toxic ions. But, in the literature at the beginning of our study we did not find information about the influence of endophytic plant-growth promoting bacteria on this process in wheat plants under Cd stress. The obtained results showed the involvement of strain 10-4 in the regulation of lignin formation, which obviously makes an important contribution to the protection by these bacteria of wheat plants from Cd-caused osmotic and oxidative cell damages, and reduced Cd accumulation in the roots and its further translocation into plants above ground parts due to the roots have a strong barrier that prevents penetration of toxic ions into cells. These observed effects certainly are of interest for further close attention and in-depth research in this direction for a more complete use of the potential of endophytic bacteria B. subtilis in environmentally oriented technologies for growing wheat, especially under adverse stress factors, including Cd toxicity.
With gratitude,
Dr. Dilara Maslennikova and co-authors

Reviewer 2 Report
Abstract: It is suggested that the abstract should include the following sections: introduction, main objective, methodology, main results and conclusions. The one presented here does not have introduction.
Use as keywords words not in title.
Title and objective do not say the same. Are you studying the effect BS or the effect of Cd stress?
Title “The Effects of Endophyte Bacillus subtilis on Growth, Tolerance and Cadmium Accumulation in Wheat under Cadmium 3 Toxicity”
Objective “This study aimed to analyze the effect of Cd stress on wheat growth, photosynthetic pigments, oxidative stress parameters, roots’ lignin content and accumulation of Cd ions in wheat plants upon application of plant growth promoting bacterial endophyte B. subtilis (BS)”
Rewrite title or objective.
Results: figures 1, 2 and 3 are in white and black, why figures 4 and 5 are in color? Does color indicate something? I suggest white and black for figures 4 and 5.
Author Response
Dear Reviewer, thank you for the attentive attitude to our article and for your comments/suggestions. We revised the manuscript according to yours and other Reviewers’ comments.
Abstract: It is suggested that the abstract should include the following sections: introduction, main objective, methodology, main results and conclusions. The one presented here does not have introduction.
Response: Introductory words were inserted in Abstract of the modified version of the manuscript.
Use as keywords words not in title.
Response: Revised.
Title and objective do not say the same. Are you studying the effect BS or the effect of Cd stress?
Title “The Effects of Endophyte Bacillus subtilis on Growth, Tolerance and Cadmium Accumulation in Wheat under Cadmium Toxicity”
Objective “This study aimed to analyze the effect of Cd stress on wheat growth, photosynthetic pigments, oxidative stress parameters, roots’ lignin content and accumulation of Cd ions in wheat plants upon application of plant growth promoting bacterial endophyte B. subtilis (BS)”
Rewrite title or objective.
Response: The title has been changed.
Results: figures 1, 2 and 3 are in white and black, why figures 4 and 5 are in color? Does color indicate something? I suggest white and black for figures 4 and 5.
Response: Revised. All figures are in black and white now.
With gratitude,
Dr. Dilara Maslennikova and co-authors

Reviewer 3 Report
The authors investigated the efficacy of endophyte Bacillus subtilis 104 (BS) on cadmium acetate (Cd) stressed Triticum aestivum L. (wheat) growth, photosynthetic pigments, oxidative stress parameters, roots’ lignin content and Cd ions accumulation in plants. The findings revealed that Cd tolerant BS improved the ability of Wheat to germinate under Cd toxicity. This study may be helpful to utilize microbial sources to counter heavy metal(s) stress on crops. The paper is generally well-written but the methodology framework is unclear and lacks potential references.
The introductory section of this article lacks critical and in-depth analysis of the available literature and comprehensive relevant work.
Comment 1: As it is a global issue, therefore, authors should incorporate quantitative data about the heavy metals toxicity in agriculture lands of different countries and it would be good to cite and relate some other studies that investigated similar aspects.
Comment 2: I suggest you add 2-3 introductory sentences about reduction of wheat yield/productivity in different regions of world. The reason to select wheat as a test crop should be explained.
Comment 3: Explain the novelty of the work in a paragraph and make a comparison with the literature. There is a major gap to link previous work and the planned objectives of this work.
Comment 4: Besides the wide use of BS, how about its contribution to native microbial communities?
Comment 5: Detailed information should be given about soil sample collection, preservation, and analytical techniques. There is no reference given in the materials and methods section
https://doi.org/10.1016/j.cej.2019.123674 https://doi.org/10.1016/j.watres.2019.03.079
https://doi.org/10.1016/j.envpol.2021.116587
Comment 7: I suggest you incorporate longitude and latitude data of sampling points and describe them in brief.
Comment 8: There are several citations in methods that should be replaced with primary references.
Comment 9: Authors should precisely focus on and strengthen this section with published research work.
Comment 10: I suggest you conduct a multivariate statistical analysis and determine the interaction effect among key parameters of this study.
Comment 11: In the discussion section, the ecological significance is not pointed out clearly by the authors.
Comment 12: In order to meet the requirements for publication, the quality of figures and tables should be improved.
Comment 13: Please rewrite this section with a prime focus on your key findings.
There are numerous low-level errors in the text (including but not limited to unit symbols, punctuation marks, font sizes, and paragraph structure…). Please check the format of the text throughout this manuscript and unified its pattern. Correcting these errors should not be the responsibility of the reviewers. Please check it carefully throughout this manuscript.
Author Response
Dear Reviewer, thank you for the attentive attitude to our article and for your comments/suggestions. We revised the manuscript according to yours and other Reviewers’ comments.
The authors investigated the efficacy of endophyte Bacillus subtilis 104 (BS) on cadmium acetate (Cd) stressed Triticum aestivum L. (wheat) growth, photosynthetic pigments, oxidative stress parameters, roots’ lignin content and Cd ions accumulation in plants. The findings revealed that Cd tolerant BS improved the ability of Wheat to germinate under Cd toxicity. This study may be helpful to utilize microbial sources to counter heavy metal(s) stress on crops. The paper is generally well-written but the methodology framework is unclear and lacks potential references.
The introductory section of this article lacks critical and in-depth analysis of the available literature and comprehensive relevant work.
Response: Revised.
Comment 1: As it is a global issue, therefore, authors should incorporate quantitative data about the heavy metals toxicity in agriculture lands of different countries and it would be good to cite and relate some other studies that investigated similar aspects.
Response: Revised. A new information added to the modified manuscript.
Comment 2: I suggest you add 2-3 introductory sentences about reduction of wheat yield/productivity in different regions of world. The reason to select wheat as a test crop should be explained.
Response: Сorrections have been made to the manuscript.
Comment 3: Explain the novelty of the work in a paragraph and make a comparison with the literature. There is a major gap to link previous work and the planned objectives of this work.
Response: Done.
Comment 4: Besides the wide use of BS, how about its contribution to native microbial communities?
Response: We did not yet analyse the effect of seed treatment with BS on native microbial communities of plants, particularly, wheat. We plan to focus on this aspect in our future works, and hopefully we will be able to present the results in our next publications.
Comment 5: Detailed information should be given about soil sample collection, preservation, and analytical techniques. There is no reference given in the materials and methods section https://doi.org/10.1016/j.cej.2019.123674 https://doi.org/10.1016/j.watres.2019.03.079
https://doi.org/10.1016/j.envpol.2021.116587
Response: The information added to the modified version of the manuscript. The strain 10-4 was isolated earlier from dryland arable soils using a classical microbiological method (Netrusov et al. 2005) and more detailed information was previously published (Lastochkina et al. 2017).
Comment 7: I suggest you incorporate longitude and latitude data of sampling points and describe them in brief.
Response: The information provided now in section 2.1.
Comment 8: There are several citations in methods that should be replaced with primary references.
Response: Done.
Comment 9: Authors should precisely focus on and strengthen this section with published research work.
Response: Revised.
Comment 10: I suggest you conduct a multivariate statistical analysis and determine the interaction effect among key parameters of this study.
Response: Done. The results presented in Subsection 3.7. of the modified manuscript.
Comment 11: In the discussion section, the ecological significance is not pointed out clearly by the authors.
Response: Appropriate information added now.
Comment 12: In order to meet the requirements for publication, the quality of figures and tables should be improved.
Response: We tried to improve the quality of figures and tables as it possible.
Comment 13: Please rewrite this section with a prime focus on your key findings.
Response: Done.
With gratitude,
Dr. Dilara Maslennikova and co-authors

Round 2
Reviewer 1 Report
In my opinion manuscript was extensively improved;
Moreover, the correlation analysis is very informative and widely improved the results part of the manuscript;
Furthermore, statement 'plant samples' in methodology was deeply explained in almost all analyses;
Unfortunately, lignin deposition documentation was weak in quality, but I appreciate that it was added - in my opinion in current version is almost the only one weak point of the manuscript;